# Radiographers' perceptions on the quality of managing general radiographic paediatric examinations through the use of a reflective tool

Kate Caruana[1]*, Chris Hayre[2], Chandra Makanjee[3]

1 Department of Medical Imaging, The Canberra Hospital, Garran, Australian Capital Territory, Australia,
2 Department of Health and Care Professions, University of Exeter, Exeter, Devon, United Kingdom,
3 Department of Medical Radiation Science, University of Canberra, Bruce, Australian Capital Territory, Australia

☯ These authors contributed equally to this work.
* kate.caruana@act.gov.au

## Abstract

### Introduction

Paediatric patients are a vulnerable population that require additional care by healthcare professionals. Quality managing these examinations ensures that effective and quality care is provided to individual patients, whilst encouraging consistency within the medical imaging department. This study explored radiographers' perspectives on quality management strategies of general radiographic paediatric examinations using a paediatric imaging reflective checklist.

### Methods

A quantitative descriptive research design with qualitative questions was used through a purposive sampling method from both public and private Australian diagnostic imaging qualified radiographers who had experience in paediatric imaging examinations. The paediatric imaging service reflective tool consisted of 65 items in total. Data analysis entailed Microsoft Excel version 16.16.6 and Jamovi version 2.3.21 for the closed-ended questions and for the open-ended responses a thematic analysis.

### Results

The participation rate was 13.2% and the most significant findings were: lead shielding was still being used at their organisation, despite recent recommendations to suspend its use; access to paediatric patient related information resources is limited; there was no involvement of families and communities regarding policy development or quality improvement measures as advocated in literature; and there was a need for enhanced specialised paediatric education, training and protocols.

**Data Availability Statement:** Data cannot be shared publicly because of the small sample size and is re-identifiable. Data are available with permission from the University of Canberra Human

Research Ethics Committee (contact via
humanethicscommittee@canberra.edu.au) for
researchers who meet the criteria for access to
confidential data.

**Funding:** The author(s) received no specific
funding for this work.

**Competing interests:** The authors have declared
that no competing interests exist.

## Conclusion

Using the paediatric patient-centred imaging reflective checklist, radiographers had an
opportunity to identify quality improvement indicators as well as issues that could further
enhance best practice principles. Further studies could inform on the validity of this reflective
tool.

## Introduction

Enhancing the quality and safety of healthcare delivery requires a strong emphasis on continu-
ous improvement of organisations and individuals, organisational support and resources as
well as systematically adjusting policies in accordance with best evidence-based practice [1, 2].
In order to accomplish this, it is crucial to identify the barriers and factors that impact the
quality of care delivered within healthcare services [2, 3]. There is an ongoing plea to enhance
the quality of medical imaging services that have expanded in resources as a result of the devel-
opment and availability of technology, resulting in an increased use of ionising radiation-
based examinations [2, 4, 5]. A quality management system should be adopted within medical
imaging departments to improve task efficiency and safety, providing quality services whilst
maximising effective care and patient satisfaction whilst maintaining appropriate radiation
doses [6, 7].

Unlike their adult counterparts, paediatric patients are a vulnerable population and present
with their own challenges to the medical imaging department, whereby efforts need to be
made to provide effective and quality services with a great attention to childhood specific dis-
eases and potential risks [8, 9]. The fundamental challenge of imaging paediatric patients lies
in determining the appropriate radiation dose, whilst maintaining acceptable image quality
[10]. In addition, radiographers are expected to perform complex tasks whilst maintaining a
high standard and sustaining quality and safety for patients which includes the paediatric pop-
ulation in a contemporary context [11, 12]. Thus, the input and skill of radiographers plays a
crucial role in justification and optimisation of radiation dose and the achievement of diagnos-
tic images, which is pivotal for the treatment and management of paediatric health conditions
[10, 13]. Radiographers are, however, limited in advancing their knowledge in paediatric imag-
ing with their currently being no postgraduate qualification in Australia [14]. Literature also
acknowledges the dearth of recognition regarding paediatric imaging as a specialty or profes-
sional registration for paediatric radiographers [14, 15].

Achieving a high level of quality imaging services requires consideration of objective and
subjective factors by radiographers, as well as the involvement and collaboration of multiple
stakeholders within and external to the organisation [9, 16]. It entails aspects of the medical
imaging procedure being considered such as patient access to the department, assessing the
child's needs, staff-related elements, and the performance of the examination [17]. Quality
managing medical imaging examinations allows inconsistencies amongst organisations, for
example, human or equipment factors, to be mitigated and controlled whilst providing
patient-centred care [6, 7]. Such variations include differing needs for each child, as well as
organisational challenges including child-appropriate protocols, specialised equipment and
environment, appropriate radiation protection and quality management [9].

There is a plethora of literature offering in-depth knowledge regarding safety culture, which
entails areas such as radiation safety, effective communication and adopting a paediatric-cen-
tred approach. Such research entails aspects of the importance of radiation protection and
dose to paediatric patients, as well as delivering paediatric specific care and communication to
provide better healthcare outcomes for children [5, 10, 18, 19]. This literature also delves into

the advantages of quality programs and the essential characteristics for their success [2, 6, 7, 10, 11]. However, there is a lack of information in relation to the quality management of paediatric imaging [2, 6], with there being no research that investigates all aspects of general radiographic paediatric examinations within a single study. Furthermore, there is no study that allows radiographers to reflect on such practice and identify areas of improvements to enhance the paediatric service.

In order to be consistent amongst individual radiographers and organisations, we must reflect on the quality of managing paediatric examinations [20]. Such reflection allows healthcare professionals to learn from experiences as well as identify areas of improvements through critical thinking, self-awareness and evaluation [21, 22]. The aim of this study is to establish radiographers' perspectives on the quality of paediatric medical imaging services using a paediatric medical imaging-centred reflective tool.

## Materials and methods

A quantitative research design with qualitative aspects was adopted for this study to acquire radiographers' insights into quality managing paediatric imaging services. A purposive sampling technique was used to recruit qualified diagnostic radiographers with experience in general radiographic paediatric examinations, involving the assistance of study participants in obtaining additional participants [23]. This sampling method was used to target eligible participants whilst aiding the researcher with recruitment. Ethics approval was received from the Faculty of Health Research Ethics Committee (HREC), University of Canberra (HREC 4780), whereby this study is a part of a larger project investigating paediatric imaging [24]. The data collection process began in July 2022 and ended in March 2023. Written or verbal informed consent was obtained from participants who agreed to participate in the study. The content of the consent form was approved by the HREC prior to dissemination of the questionnaire. If verbal consent was given, documentation was included in the researcher's notes.

The paediatric-centred reflective tool was developed from a published person-centred reflective tool [25]. The first section entailed 15 items on participant demographic and professional background information. Section two required respondents to indicate "Yes" or "No" and a comment section on aspects of access to services, efficiency of task performance, patient care and wellbeing, ethical considerations, staff related elements and quality management. The "Yes / No" responses allowed an understanding of general radiographic paediatric examinations to be obtained [26], whereby radiographers reflected on all aspects of these examinations. Open-ended questions were also used in section two, providing an option for participants to identify new issues not captured in the closed-ended questions [27]. This consisted of a "Recommended action plan" and "Other comments" section for each item, which allowed for participants to outline what is currently being performed at their organisation as well as to provide additional suggestions or comments for further improvement [25, 27].

To ensure content validity, the research instrument was piloted by two clinical radiographers and two academic radiographers [23, 28]. The feedback was incorporated into the study prior to its dissemination. Internal consistency of dichotomous data was determined by the use of McDonald's omega [29, 30], as depicted in Table 1. Similar to Cronbach alpha, McDonald's omega is expressed as a number between 0 and 1, with optimal values ranging between 0.7 and 0.95, thus, indicating internal consistency of the instrument [29, 31, 32].

## Results

In this section, the demographic data, followed by the checklist item responses and thematic analysis is depicted. An overall response rate was 13.2% (n = 60) of the 453 potential

**Table 1. McDonald's omega scores for each section.**

| Sections from tool | McDonald's omega | Number of items | Acceptance reason |
|---|---|---|---|
| Ease of access to services and the service delivery | 0.615 | 5 | Poor internal consistency but acceptable due variations of participants perceptions |
| Efficiency of task performance | 0.798 | 11 | Moderate internal consistency |
| Patient care and wellbeing (include family and caregiver) | 0.688 | 7 | Poor internal consistency but acceptable due variations of participants perceptions |
| Ethical consideration | 0.934 | 3 | Excellent internal consistency |
| Staff related elements | 0.711 | 8 | Moderate internal consistency |
| Quality management | 0.749 | 10 | Moderate internal consistency |

The quantitative data was analysed using Microsoft Excel version 16.16.6 and Jamovi version 2.3.21. The qualitative data was analysed thematically.

participants recruited for this study. Dedicated paediatric organisations were targeted, with only 4 responses provided and as a result, non-paediatric organisations were approached which consisted of the 56 remaining participants. Additionally, 60% were from the Australian Capital Territory, with 35% of participants from other states across Australia and the remaining 5% identifying two different locations which indicated multiple places of employment. The majority of participants (80%) were employed in a metropolitan institution and 18% at a regional/rural area, with 70% in the public sector and 25% employed at a private organisation. Their employment status varied, 61% being full-time, 15% part time and 7% employed on a casual basis. The age range was between 21 to 66 years and the mean age was 31.3 years. The majority of participants (75%) had an undergraduate bachelor's degree, few (18%) had a master's qualification and none with a specialised paediatric qualification. Slightly higher than half of the participants (55%) had $\leq$ 4 years' experience in paediatric imaging and the remaining varied between 5 to 46 years in experience.

As depicted in Table 2, ease of access to services was positively rated for the majority, except resources of information at approximately 58%. Efficiency of the task performance was exceedingly high, whereas safety in terms of radiation protective gear was approximately 66%. Patient care and wellbeing and ethical considerations received high ratings by participants, with the exception of cognitive assessment which received approximately 66%. In terms of the staff related elements, over 80% participants stated "Yes" to both paediatric specific equipment and patient satisfaction. Availability of educational support was 60%, however, professional development of staff was positively rated. The majority of participants (75%) felt there was multidisciplinary team engagement and peer debriefing. Regarding quality management, more than 75% stated there was stakeholder involvement in quality improvement measures as well as quality assurance and quality control protocols in place alongside dedicated imaging protocols. However, family and community engagement in quality management and paediatric service delivery was average (50%).

## Evaluation of medical imaging services

Some participants identified gaps in terms of access to parking, including aspects of navigating as illustrated in the following quotes:

"*Parking close to the hospital often unavailable during business hours around the hospital*" (P2).

"*Poor parking and signage located on site*" (P13).

**Table 2. Radiographers' responses to quality indicators.**

| Category | Items | Yes (%) | No (%) | Response rate (%) |
|---|---|---|---|---|
| Ease of access to services and the service delivery | Resources for information (e.g. website to inform patient decisions on practice or examinations; patients access to education programmes and information leaflets). | 58.3 | 41.7 | 100.0 |
| | Appointment scheduling and emergency/urgent access to imaging. | 98.3 | 1.7 | 100.0 |
| | Navigation from home to point of face-to-face contact to the radiology department (e.g. parking, signage, web information). | 81.7 | 15.0 | 96.7 |
| | Documentation and verification/record keeping (request orders are appropriately completed and filed, access to old imaging examinations and reports). a. Within the radiology department. | 95.0 | 3.3 | 98.3 |
| | b. Other healthcare professionals (e.g. means to obtain and share previous and current imaging with each part of the patient healthcare team such as nurses, referring doctor, etc.). | 86.7 | 10.0 | 96.7 |
| Efficiency of task performance | Pre-examination a. Appropriateness and quality of request. | 83.3 | 13.3 | 96.6 |
| | b. Patient identification verified. | 98.3 | 0.0 | 98.3 |
| | c. Explained examination to patient/parent. | 96.7 | 3.3 | 100.0 |
| | d. Verified pregnancy/last menstrual period (LMP), if appropriate. | 95.0 | 3.3 | 98.3 |
| | Examination a. Beam body part image receptor aligned (with or without grid use). | 100.0 | 0.0 | 100.0 |
| | b. SID checked. | 88.3 | 10.0 | 98.3 |
| | c. Patient body part positioned and collimation applied. | 96.7 | 3.3 | 100.0 |
| | d. Anatomical marker placement. | 85.0 | 13.3 | 98.3 |
| | e. Shielding, where applicable. | 66.7 | 33.3 | 100.0 |
| | f. Appropriate application of immobilisation aid, where applicable (e.g. physical, chemical, mechanical, psychological). | 95.0 | 3.3 | 98.3 |
| | g. Exposure technique selection. | 96.7 | 1.7 | 98.4 |
| | h. Final adjustment if necessary (e.g. rotation, tilt, etc.). | 96.7 | 1.7 | 98.4 |
| | i. Exposure taken. | 98.3 | 0.0 | 98.3 |
| | Post examination a. Processing and image review (e.g. acceptable range of image quality, artifacts). | 100.0 | 0.0 | 100.0 |
| | b. Decision to repeat or proceed to archiving and ready for reporting. | 96.7 | 3.3 | 100.0 |
| Patient care and wellbeing (include family and caregiver) | Care communication and interactions (issues of literacy, language proficiency, physical interaction, touching/visual stimuli, how the child communicates). | 96.7 | 3.3 | 100.0 |
| | Child's interests (e.g. to assess what works and could calm the child, check reaction to sound, light). | 91.7 | 8.3 | 100.0 |
| | Assessing cognitive impairment and stage of development (Sensitometer/preoperational stage/concrete operational stage/formal operational stage). | 66.7 | 30.0 | 96.7 |
| | Socio-cultural and emotional considerations. | 91.7 | 6.7 | 98.4 |
| | Power relations between the patient, accompanying adults and radiographer (e.g. radiographer is in a more powerful position than the patient). | 80.0 | 13.3 | 93.3 |
| | Space/opportunity for decisions, discussions, consent, information exchange with patient to allow an understanding of the patient's needs. | 95.0 | 5.0 | 100.0 |
| | Behaviour of staff in an appropriate demeanour. | 98.3 | 1.7 | 100.0 |
| Ethical consideration | Child's rights/risks versus harm/benefits. | 98.3 | 0.0 | 98.3 |
| | Ethical and legal guidelines for the care of adolescents regarding privacy and medical decision-making. | 96.7 | 3.3 | 100.0 |
| | Consent consideration and assent based on the child's age and developmental understanding. | 96.7 | 1.7 | 98.4 |
| | Guidelines for appropriate age and sex of the patient, especially considering adults who may be accompanying minors. | 90.0 | 3.3 | 93.3 |

(*Continued*)

**Table 2.** (Continued)

| Category | Items | Yes (%) | No (%) | Response rate (%) |
|---|---|---|---|---|
| Staff related elements | Paediatric medical imaging equipment and accessories is compliant (e.g. equipment size, space and appropriate function status). | 83.3 | 13.3 | 96.6 |
| | Mechanism of reporting errors (e.g. incident writing). | 98.3 | 1.7 | 100.0 |
| | Patient data utilisation (e.g. reject-film analysis, ordering of equipment). | 90.0 | 8.3 | 98.3 |
| | Evaluation of patient, family and or caregiver satisfaction (e.g. professionalism, pacing the examination, efficiency of task performance, equitable treatment). | 83.3 | 15.0 | 98.3 |
| | Education programmes for patients, students and staff. | 60.0 | 38.3 | 98.3 |
| | Peer debriefing. | 75.0 | 23.3 | 98.3 |
| | Professional development of staff. | 93.3 | 6.7 | 100.0 |
| | Multidisciplinary team engagement and regular review throughout the institution for paediatric medical imaging (e.g. including occupational health and safety). | 70.0 | 25.0 | 95.0 |
| Quality management | Safety (cultural, infection, radiation safety, contrast agents, communication, safe patient environment, incident reporting which could include patient handling as well as radiation dose incidents). | 100.0 | 0.0 | 100.0 |
| | Stakeholder participation in relation to quality improvement measures (internal stakeholders that operate within the institution such as management team, employees, patients and external stakeholders that operate outside the institution such as external funders, professional bodies, accrediting bodies). | 76.7 | 16.7 | 93.4 |
| | Institutional culture on safety management (e.g. error disclosure and risk management). | 96.7 | 1.7 | 98.4 |
| | Monitoring of quality improvement (e.g. through quality assurance, quality control of equipment and workplace with paediatric focus). | 80.0 | 20.0 | 100.0 |
| | Standards and compliance with authorising institutional and national regulatory authorities. | 91.7 | 8.3 | 100.0 |
| | Paediatric medical imaging procedure protocols, policies and guidelines. a. According to developmental needs (e.g. Skeletal survey accidental and non-accidental injuries). | 96.7 | 3.3 | 100.0 |
| | b. Dose reference levels. | 88.3 | 10.0 | 98.3 |
| | c. Unique needs of paediatric patients. | 91.7 | 8.3 | 100.0 |
| | d. Timely advanced level of care. | 91.7 | 6.7 | 98.4 |
| | e. Internal and external institutional referral for efficient safe transitioning. | 81.7 | 10.0 | 91.7 |
| | Involving families and the community in quality assurance, policy development and facility improvements. | 50.0 | 41.7 | 91.7 |

With reference to Table 2, participants comments throughout the various categories were captured in three themes which consist of the following: evaluation of medical imaging services; issues pertaining to best practice principles and quality assurance; and support mechanisms and resources.

"*We have poor signage to get to medical imaging*" (P8).

Participants (42%) stated "No" in relation to information resources for the dichotomous question, indicating there was a scarce amount of information available. Examples consisted of:

"*Resources are limited, only a website is available displaying a two minute generic video for paediatric x-rays*" (P5) and in some instances

"*. . . there is no website, education programmes or information leaflets available to patients regarding paediatric examinations*" (P31).

This limitation could be overcome by:

"*Having leaflets both physical and online to provide patient information on the risks and benefits associated with paediatric imaging. QR [quick response] code within examination rooms for parents/guardians to access this information*" (P7).

"*. . . has developed an app ("Okee") for this exact purpose. The app can be found through various app stores*" (P21).

"*Fact sheets on simple dose metrics provided in the waiting room for patients*" (P2).

## Issues pertaining to best practice principles and quality assurance

Despite the majority indicating that medical imaging equipment was tailored for paediatric examinations, a few participants indicated otherwise.

"*None of the sites that I work at have equipment or accessories specific to paediatric patients*" (P31).

"*There isn't a paediatric focus on monitoring of quality improvement, in regard to equipment*" (P5).

"*Low levels of paediatric focus in these area*" (P6).

Participant 8 outlined that a potential reason for this could be due to financial constraints.

"*Financial analysis prevents us from getting funding for specific paediatric equipment as we don't have a specialised paediatric department and such we don't get many patients*" (P8).

Although the majority of participants stated "Yes" in relation to efficiency of task performance, the collimation applied to the body part region of interest in part of radiation safety mechanism was of concern as stated:

"*Sometimes collimation is kept wider to ensure anatomy required is captured on a moving or struggling child*" (P8).

"*Often left slightly open, in case of pt [patient] movement*" (P24).

Similarly, application of lead protective shielding appears to be discretionary as shared by Participants 5 and 9.

"*Used when appropriate*" (P5).

"*Paediatric shielding used if appropriate*" (P9).

Though most participants had knowledge on dose reference levels, some participants (7%) identified there are no such charts available for paediatric general x-ray at their organisation, as stated by the following participants.

"*For CT [computed tomography] yes. Not x-ray*" (P11).

"*There are no charts or information sheets regarding dose reference levels for paediatric examinations at the company*" (P31).

Specifically, one participant acknowledged that these charts "*Don't exist in Australia for children's x-rays*" (P39).

Surveys for patients and families, are valuable because feedback can be used *". . . to integrate changes to the department as they are the consumers"* (P30). In this regard, Participant 31 commented:

*". . . I believe the company has surveys at reception available for all patients (paediatric or not) to give feedback about their experience"* (P31).

Some participants stated that to strengthen feedback *"greater access to feedback forms in reception for pts [patients] to fill out post exam"* (P14) and *"More satisfaction surveys to be completed"* (P49).

Few participants also indicated patient, family and or community involvement was inappropriate because *". . . it is not their specialty/job?"* (P37) and *". . . do not understand much of these processes"* (P21).

On the other hand, some participants like Participant 31 stated:

*"Improvements are always a good thing, and I do think that families and the community should be included in those conversations"* (P31).

## Support mechanisms and resources

Regarding professional development and educational support, participants were *". . . not aware of any company led education programmes for staff or patients"* (P31) and that was *"limited explicit paediatric content"* (P6). If there were opportunities for education and/or professional development, it was *"limited for pts [patients] & [and] students"* (P18). Whereas others indicated that staff educational opportunities were *"limited in regard to paediatric imaging"* (P2) or *"not specific to paed's [paediatrics]"* (P4).

Participants identified the need for an increased amount and improved education with regards to programmes and professional development of staff.

*". . . exist but could be much more/improved"* (P12).

For example,

*"Need more CPD [continuing professional development] specific to imaging paeds [paediatrics]"* (P43).

Similarly, another participant stated:

*"More available paed [paediatric] specific CPD [continuing professional development] would be helpful"* (P11).

Participants also expressed a greater desire for education in specific areas. These comments included:

*". . . regular education regarding patient communication"* (P21),

*"More care/training for children with disabilities"* (P19) and

**"*. . . cultural awareness"* (P21).**

## Discussion

Reflection is an important process for understanding quality assurance and improvement strategies and implementation thereof [33]. Based on the undertaking, a reflection on the quality of paediatric imaging services revealed interesting results.

Organisations must seek feedback from patients and families to identify areas of improvement and enhance the radiology service and quality [34, 35]. However, the results from the study conducted found a discrepancy amongst participants relating to the availability and completion rate of patient and family surveys. Similarly, Lee et al. [36] reported feedback forms were available, but organisations need to improve the diversity and accessibility of these to enhance, monitor and assure the quality of care. On an analogous note, Berger et al. [37] found a myriad of patient feedback forms were available, which need to be promoted and patients encouraged to complete. In addition to organisations seeking feedback from patients and families, they should be involved in the development and implementation of policies to improve safety and care [38]. It was identified, however, that participants from the study were uncertain in relation to the families and patient involvement in such policy development. Literature reinforces the importance such a perspective can provide and acknowledges that stakeholders should be involved and work collaboratively with service providers for the development of policies and quality improvement programs, leading to improved outcomes and better quality and safe health care for patients [39].

Access to medical imaging departments includes appropriate parking and signage that is clear and indicates the location of the department [17, 40]. For patients, the implications of insufficient parking are creating stress and anxiety [17]. The study conducted found such services was insufficient, which could impact the quality of the patient's healthcare experience.

Research identifies the benefit of having specific paediatric medical imaging equipment and quality assurance programs implemented into organisations, enhancing the service delivery and care paediatric patients receive [11, 41]. In this study, most radiographers reported the equipment to be suitable for paediatric examination, however, there were no dedicated paediatric equipment and quality assurance for this specific population group. Similar to a study by Billinger et al. [42], there was a significant variation of dose reference levels between different hospitals. Currently, "There are currently no Australian national Diagnostic Reference Levels published for paediatric, or adult, general radiography examinations" [43].

There is a notable absence of specialised paediatric imaging postgraduate qualifications among the radiography participants, aligning with literature [14, 15, 24]. Contributing factors could be insufficient avenues or motivations for specialised paediatric radiographers as well as the limited recognition by regulatory boards identifying paediatric as a distinct sub speciality within medical imaging [15, 24]. Quality health care is defined as "the degree to which health services for individuals and populations increase the likelihood of desired health outcomes and are consistent with current professional knowledge" [44]. Therefore, education and training on contemporary paediatric imaging processes and procedures is paramount [14]. The results from the study are consistent with previous research like Alsleem and others which reported a lack of dedicated paediatric education programmes [45]. Likewise, Ewuzie concludes that a lack of these programmes could hamper the services offered to paediatric patients and must be improved [46].

An issue highlighted in this study is immobilisation aids and current best-practice principles. Christie et al. also found more than 30% of radiographers believed their training in the use of immobilisation aids for paediatric patients was poor or lacking [47]. Similarly, Satharasinghe et al. found radiographers were tending to use a larger than required x-ray field in an attempt to avoid repeating the imaging examination [48]. However, unnecessary radiation

exposure to the patient can result from inadequate collimation and thus, radiographers must improve such skills to improve patient outcomes [49, 50]. In terms of using lead shielding, Cardoso et al. found uncertainty among radiographers, specifically about gonad contact shielding [51]. In this study, the use of lead shielding was discretionary. A suggestion to resolve this issue could be to review and update recent guidelines regarding lead shielding within organisations [52]. On the contrary, regulatory bodies advocate discontinuation of shielding device [53, 54]. Such use of shielding provides no significant advantage and can increase patient radiation exposure, potentially compromising the effectiveness of the imaging [55].

In literature, communication is a major component in healthcare related contexts in terms of confidence and skills in ensuring benefit-risk information is communicated clearly [56, 57]. In order to achieve an effective exchange of information, compassion, recognition, acknowledgement and unprejudiced acceptance of diversity must occur [56]. An interesting finding were barriers faced with paediatric patients with intellectual disabilities [57, 58]. Enhancing radiographer's education and skill to ensure they are better equipped to address these barriers would improve the paediatric radiography services offered by organisations [46, 57, 58]. In this study, participants also expressed a desire for improved cultural competence, allowing individual patient needs to be addressed and reduce health disparities amongst diverse populations [59, 60].

Another aspect of communication entails patients desires for written resources of information to be made available to them regarding paediatric imaging, such as rationale for ordering specific imaging examination [61]. This could include hospital-endorsed internet sites and take-away printed material to retain and use for reference [61]. However, Bray et al. reported children were frequently not given or assisted in accessing healthcare information, despite the desire to do so prior to a medical procedure [62].

A unique contribution of this study is the clinically based paediatric-centred quality management reflective tool. The findings of this study could be used as a baseline for a similar study of this nature, including validation of the effectiveness of the reflective tool as a part of a quality improvement strategy within a clinical setting. This would aid in identifying strengths and weaknesses in the delivery of paediatric medical imaging services. The outcomes inform and contribute towards existing quality improvement endeavours at an organisational level, such as, devising appropriate action plans, timelines, and/or strategies. For example, prioritising the professional development of staff in paediatric specific education and communication aspects as well as investing in dedicated paediatric equipment and instrumentation. By addressing these implications, healthcare organisations can improve and ensure a high quality of care services, thereby contributing to better health outcomes for paediatric patients.

Despite the strengths of the study, there are some limitations. Specifically, the low participant response rate was as a result of professionals experiencing burnout due to post COVID-19 fatigue, high workplace demands and the eligibility criteria to participate in the study. This limits the generalisability of the results. Additionally, most of the participants had less than 4 years' experience and not all states were included. Future studies could engage a larger population through use of a medical imaging team-based approach and involve practice managers, quality and safety management officers and target medical imaging practices across all states.

## Conclusion

This tool enabled participants to reflect on their own practice and provided valuable insights into the challenges and opportunities to enhance the paediatric imaging services at an individual and organisational level. It emphasises the importance of addressing issues associated with equipment, quality assurance and best practice principles. Additionally, communication,

patient involvement and educational support must be improved to enhance the quality of care provided to paediatric patients. However, these results lack generalisability due to the restricted response rate. Further research and action are needed to address these identified gaps and enhance the overall quality of paediatric radiology services.

## Supporting information

**S1 Checklist. A quality management checklist for paediatric imaging examinations.**
(PDF)

## Acknowledgments

The authors thank the radiographers who participated in the study.

## Author Contributions

**Conceptualization:** Kate Caruana, Chandra Makanjee.

**Formal analysis:** Kate Caruana.

**Investigation:** Kate Caruana.

**Methodology:** Kate Caruana.

**Project administration:** Kate Caruana.

**Resources:** Chandra Makanjee.

**Supervision:** Chandra Makanjee.

**Writing – original draft:** Kate Caruana, Chris Hayre, Chandra Makanjee.

**Writing – review & editing:** Kate Caruana, Chris Hayre, Chandra Makanjee.

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
