## [Decision Letter · Decision Letter 0]

25 Oct 2023

PONE-D-23-26933Radiographers’ perceptions on the quality of managing general radiographic paediatric examinations through the use of a reflective toolPLOS ONE

Dear Dr. Caruana,

Thank you for submitting your manuscript to PLOS ONE. After careful consideration, we feel that it has merit but does not fully meet PLOS ONE’s publication criteria as it currently stands. Therefore, we invite you to submit a revised version of the manuscript that addresses the points raised during the review process.

The paper is addressing an important area and it would be good for the readership of the journal. Before it can be published, the authors should clearly highlight the gap in literature that the study sought to address and also discuss the key implications of the findings to practice. 

We look forward to receiving your revised manuscript.

Kind regards,

Aloysius Gonzaga Mubuuke

Academic Editor

PLOS ONE

Journal Requirements:

Additional Editor Comments:

The authors should try to highlight the gap in literature that the study sought to address in the introduction and they should also discuss the implications for practice in the discussion

Reviewers' comments:

Reviewer's Responses to Questions

**Comments to the Author**

1. Is the manuscript technically sound, and do the data support the conclusions?

Reviewer #1: Yes

2. Has the statistical analysis been performed appropriately and rigorously? 

Reviewer #1: Yes

3. Have the authors made all data underlying the findings in their manuscript fully available?

Reviewer #1: Yes

4. Is the manuscript presented in an intelligible fashion and written in standard English?

Reviewer #1: Yes

5. Review Comments to the Author

Reviewer #1: The authors discuss the reflective tool in the Materials and techniques section. Even while reflection is addressed, it is unclear where the qualitative technique fits in. The author should be more precise in defining and specifying the inclusion of the qualitative component in their study by showing that 'open-ended questions.' were used in section two.

The response rate was quite low. The author should include brief details for the reasons.

The conclusion of the study could be improved to summarise the highlights of the study better. The current conclusion does not summarise the findings well. The authors could indicate non-generalisability of the findings more so, because of the response rate. Also, indicate areas for future research and the importance of using the tool.

6. PLOS authors have the option to publish the peer review history of their article (what does this mean?). If published, this will include your full peer review and any attached files.

Reviewer #1: No

---

## [Author Response · Author response to Decision Letter 0]

16 Nov 2023

Rebuttal Letter

Original Manuscript ID: PONE-D-23-26933

Original Article Title: “Radiographers’ perceptions on the quality of managing general radiographic paediatric examinations through the use of a reflective tool”

To: PLOS ONE

Re: Response to Reviewers

Dear Editor,

Thank you for allowing a resubmission of our manuscript, with an opportunity to address the reviewers’ comments.

We are uploading 

(1) Our point-to-point response to the comments below (Response to Reviewers)

(2) A marked-up copy with track changes (Revised Manuscript with Track Changes)

(3) An unmarked version without track changes (Manuscript)

Kind regards,

Kate Caruana

Corresponding Author

Medical Radiation Practitioner 

Email: kate.caruana@act.gov.au

Tel No: (+61) 498 108 772

Date: 14 November 2023

Reviewer#1, Concern #1: 

Reviewer’s comment: The authors should try to highlight the gap in literature that the study sought to address in the introduction and they should also discuss the implications for practice in the discussion.

Authors’ response: We have revised the introduction and discussion, adding some information regarding the gap of literature and implications for practice.

Reviewer#1, Concern #2: 

Reviewer’s comment: The authors discuss the reflective tool in the Materials and techniques section. Even while reflection is addressed, it is unclear where the qualitative technique fits in. The author should be more precise in defining and specifying the inclusion of the qualitative component in their study by showing that 'open-ended questions.' were used in section two.

Authors’ response: We have addressed this concern in the materials and methods section, clearly showing that open-ended questions were used in section two of the questionnaire. 

Reviewer#1, Concern #3: 

Reviewer’s comment: The response rate was quite low. The author should include brief details for the reasons.

Authors’ response: This has been previously addressed in the discussion within the limitations of the study. We have revised this section. 

Reviewer#1, Concern #4: 

Reviewer’s comment: The conclusion of the study could be improved to summarise the highlights of the study better. The current conclusion does not summarise the findings well. The authors could indicate non-generalisability of the findings more so, because of the response rate. Also, indicate areas for future research and the importance of using the tool.

Authors’ response: Many thanks for your kind suggestion. We have rigorously revised the conclusion, summarising the key findings as well as providing information regarding future research. 

NOTE: We have further addressed the Journal Requirements as follows:

Concern #1:

Authors’ response: We have reviewed and followed the PLOS ONE’S style and format in the revised manuscript.

Concern #2:

Authors’ response: We have amended the cover letter to address the ethical/legal restrictions of sharing the data set.

Concern #3:

Authors’ response: We have reviewed the reference list and ensured its correctness and completeness. We have added and amended the reference list with additional papers to enhance the literature gap as requested by the reviewer. Whilst no papers were removed/retracted from the reference list as all papers were available, one reference was updated to reflect the latest best practice.

---

## [Editor Report · Decision Letter 1]

27 Nov 2023

Radiographers’ perceptions on the quality of managing general radiographic paediatric examinations through the use of a reflective tool

PONE-D-23-26933R1

Dear Dr. Caruana,

We’re pleased to inform you that your manuscript has been judged scientifically suitable for publication and will be formally accepted for publication once it meets all outstanding technical requirements.

Kind regards,

Aloysius Gonzaga Mubuuke

Academic Editor

PLOS ONE

Additional Editor Comments (optional):

Thank you for addressing all comments
---

## [Editor Report · Acceptance letter]

29 Nov 2023

PONE-D-23-26933R1 

Radiographers’ perceptions on the quality of managing general radiographic paediatric examinations through the use of a reflective tool 

Dear Dr. Caruana:

I'm pleased to inform you that your manuscript has been deemed suitable for publication in PLOS ONE. Congratulations! Your manuscript is now with our production department. 

Kind regards, 

on behalf of

Dr. Aloysius Gonzaga Mubuuke 

Academic Editor

PLOS ONE